# TOWARDS EXACT COMPUTATION OF INDUCTIVE BIAS

## ABSTRACT

Much research in machine learning involves finding appropriate inductive biases (e.g. convolutional neural networks, momentum-based optimizers, transformers) to promote generalization on tasks. However, quantification of the amount of inductive bias associated with these architectures and hyperparameters has been limited. We propose a novel method for efficiently computing the inductive bias required for generalization on a task with a fixed training data budget; formally, this corresponds to the amount of information required to specify well-generalizing models within a specific hypothesis space of models. Our approach involves sampling from the hypothesis space and modeling the loss distribution of hypotheses to estimate the required inductive bias for a task. Unlike prior work, our method provides a direct estimate of inductive bias without using bounds and is applicable to diverse hypothesis spaces. Moreover, we derive approximation error bounds for our estimation approach in terms of the number of sampled hypotheses. Consistent with prior results, our empirical results demonstrate that higher dimensional tasks require greater inductive bias. We show that relative to other expressive model classes, neural networks as a model class encode massive amounts of inductive bias. Furthermore, our measure quantifies the relative difference in inductive bias between different neural network architectures (e.g. with varying width and depth). Our proposed inductive bias metric provides an information-theoretic interpretation of the benefits of specific model architectures for certain tasks and provides a quantitative guide to developing tasks requiring greater inductive bias, thereby encouraging the development of more powerful inductive biases.

## 1 INTRODUCTION

Generalization is a fundamental challenge in machine learning, as models must be able to perform well on unseen data after being trained on a limited set of examples. To achieve this, researchers have extensively studied the role of inductive biases, which are prior assumptions or restrictions embedded within learning algorithms, in promoting generalization. These biases can take various forms, such as architectural choices (e.g., convolutional neural networks, momentum-based optimizers, transformers) or hyperparameter settings, and they shape the space of hypotheses that the model can consider.

Despite the importance of inductive biases, quantifying the amount of bias associated with different architectural and hyperparameter choices has remained challenging. Inductive bias can be formulated as the amount of information required to specify well-generalizing models within a *hypothesis space* of models (Chollet, 2019; Boopathy et al., 2023). Previous attempts at measuring inductive bias have often provided only upper bounds or have been limited to specific model classes. This limitation hinders a comprehensive understanding of how different biases contribute to generalization and impedes the systematic development of more effective biases.

In this paper, we propose a novel and efficient method for computing the inductive bias required for generalization on a task under fixed training data budget. Unlike prior work, our approach provides a direct estimate of inductive bias without relying on bounds. Moreover, it is applicable to diverse hypothesis spaces, allowing the computation of inductive bias *within* the context of particular model classes such as neural networks. We believe more precise and flexible computation of inductive bias is practically valuable:

First, by quantifying the amount of inductive bias associated with different architectural choices, researchers can gain profound insights into how specific design decisions affect the model's ability

to generalize. This understanding helps identify which architectural features contribute most significantly to improved performance and informs the development of more tailored and task-specific models. Armed with a quantitative measure of inductive bias, practitioners can make more informed decisions about which architectural choices to prioritize when building and optimizing machine learning models. This, in turn, can lead to more efficient model development processes and improved real-world applications.

Second, our inductive bias measure serves as a practical guide for designing tasks that demand higher levels of inductive bias. By precisely estimating the amount of inductive bias needed for a given task, researchers can intentionally craft benchmarks that challenge the boundaries of generalizability of current models. This approach encourages the development of more powerful model architectures and learning algorithms, fostering innovation in the field.

We summarize our contributions as follows:

- We propose a definition of inductive bias with an explicit dependence on the hypothesis space within which models are defined.
- We develop an efficient sampling-based algorithm to compute the inductive bias required to generalize on a task. Unlike prior work, the method can be applied to parametric and non-parametric hypothesis spaces.
- We derive an upper bound on the approximation error of inductive bias estimate; the approximation error scales inversely with the number of sampled hypotheses.
- We empirically apply our inductive bias metric to a range of domains including supervised image classification, reinforcement learning (RL) and few-shot meta-learning. Consistent with prior work, we find that higher dimensional tasks require greater inductive bias.
- We empirically find that neural networks encode massive amounts of inductive bias relative to other expressive model classes. Furthermore, we quantify the difference in inductive bias provided by different neural network architectures within a neural network hypothesis space.

## 2 RELATED WORK

**Generalization vs. Sample Complexity**    Traditionally, the generalizability of machine learning models has been analyzed in terms of sample complexity, which is the amount of training data required to generalize on a task (Cortes et al., 1994; Murata et al., 1992; Amari, 1993; Hestness et al., 2017). Measures such as Rademacher complexity (Koltchinskii & Panchenko, 2000) and VC dimension (Blumer et al., 1989) quantify the capacity of a model class and provide upper bounds on sample complexity, with less expressive model classes requiring fewer samples. More recently, data-dependent generalization bounds have been proposed, yielding tighter bounds based on dataset properties (Negrea et al., 2019; Raginsky et al., 2016; Kawaguchi et al., 2022; Lei et al., 2015; Jiang et al., 2021). Additionally, scaling laws for neural networks have modeled learning as kernel regression, revealing that sample complexity scales exponentially with the intrinsic dimensionality of data (Bahri et al., 2021; Hutter, 2021; Sharma & Kaplan, 2022).

**Generalization vs. Inductive Bias Complexity**    The importance of inductive biases in promoting generalization has been widely recognized, starting with the No Free Lunch theorem (Wolpert, 1996) which states that no learning algorithm can perform well on *all* possible tasks: learning algorithms require inductive biases tailored to specific sets of tasks. Subsequent studies have further emphasized the role of inductive biases in learning (Hernández-Orallo, 2016; Haussler, 1988; Du et al., 2018; Li et al., 2021), showing that specific abilities, biases, and model architectures contribute to the prior knowledge that facilitates generalization. Despite the central role of inductive biases, work on quantifying them has been limited. Chollet (2019) proposes measuring the generalization difficulty of a task as the amount of inductive bias required for a learning system to perform the task in addition to any training data provided. Boopathy et al. (2023) provides an upper bound on the *inductive bias complexity* of a task (i.e. how much inductive bias is required to generalize on a task) based on task properties. In particular, it finds that higher-dimensional tasks require exponentially greater inductive bias, mirroring results for sample complexity. In this work, we aim to more precisely and directly estimate the required inductive bias of a task without the use of bounds. Moreover, unlike Boopathy et al. (2023), our approach can compute inductive bias complexity within general hypothesis spaces: it

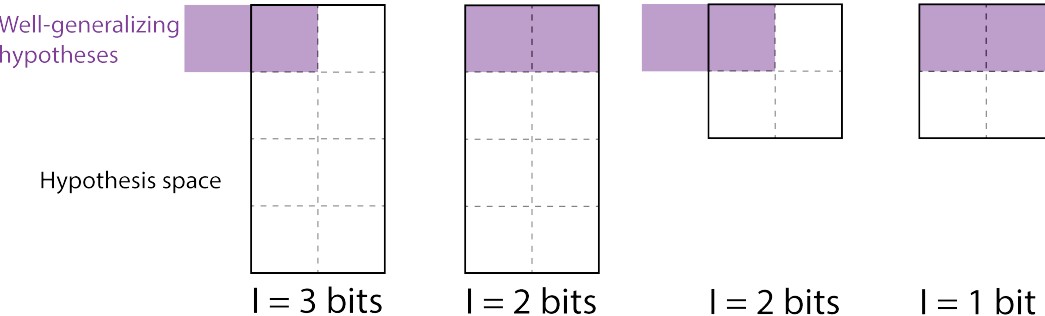

Figure 1: Illustration of how the required inductive bias for a task can be computed from the hypothesis space and the region of well-generalizing hypotheses. Black boxes indicate hypothesis spaces; $p_h$ is a uniform distribution over each box. Purple indicates regions of well-generalizing hypotheses. Inductive bias is the negative log of the fraction of hypothesis space that generalizes well; it depends on both the size of the hypothesis space as well as how much the hypothesis space overlaps with well-generalizing hypotheses. Different hypothesis spaces may yield different inductive bias estimates *even on the same task* (i.e. the same set of well-generalizing hypotheses).

allows for *context-specific* computation of inductive bias (e.g. the inductive bias required to generalize on ImageNet (Deng et al., 2009) classification *assuming* models are convolutional neural networks).

## 3 QUANTIFYING INDUCTIVE BIAS

In this section, we first provide a formal quantitative definition of the inductive bias of a model class. We then propose a method of efficiently sampling models from the model class. Given the sampled models, we then estimate the distribution of test set errors of the models; this allows us to efficiently and precisely estimate inductive bias. We prove that the approximation error of our algorithm can be bounded in terms of the number of sampled models.

### 3.1 DEFINITION OF INDUCTIVE BIAS

Intuitively, inductive biases are any properties of a model class that enhance generalization on a task. Boopathy et al. (2023) proposes quantifying the amount of inductive bias required to generalize on a task based on the probability that a model that fits a training set also generalizes to a test set. Importantly, this definition assumes that there exists a *hypothesis space* in which all models are contained; training data and inductive biases help winnow this space to a well-generalizing set of hypotheses.

We first clarify the relationship between a hypothesis space, models, and model classes. A hypothesis space is a set of all possible models that could be used to solve a given problem. A model is a specific hypothesis from a hypothesis space. A model class is a set of models (i.e. a subset of the hypothesis space) typically selected by a model designer to include mostly well-generalizing models. Inductive bias is quantified as the amount of information required to specify a well-generalizing model class within a hypothesis space.

Note that the size of the hypothesis space can strongly affect the magnitude of the inductive bias, but in Boopathy et al. (2023) the dependence on the hypothesis is implicit. Here we formally define inductive bias in a similar manner to Boopathy et al. (2023) but provide a way to explicitly include the distribution of models in the hypothesis space. The method of Boopathy et al. (2023) was task-dependent but relatively model-independent given a very broad hypothesis space. Allowing for an explicit dependence on hypothesis space allows us to compute inductive bias for a variety of more narrowly defined hypothesis families (e.g. neural networks versus Gaussian RBF models; or more finely, different neural network architectures), across a variety of domains ranging from supervised classification to RL as we will empirically show. Next, we present our formal definition of amount of inductive bias:

**Definition 1.** Let $\mathcal{H}$ be a set of hypotheses, and let hypothesis distribution $p_h$ define a probability distribution over these hypotheses. Suppose there exists a loss function $L$ that maps a hypothesis $h \in \mathcal{H}$ and a task input $x \in \mathcal{X}$ to a scalar: $L : \mathcal{H} \times \mathcal{X} \to \mathbb{R}$. Finally, suppose there exists a test distribution $p_x$ over the task inputs. With respect to distribution $p_h$, the amount of inductive bias required to achieve test set error rate $\varepsilon$ on a task is:

$$I(\varepsilon, p_h, p_x, L) = -\log \int \mathbf{1}(\mathbb{E}_{x \sim p_x}[L(h, x)] \leq \varepsilon) p_h(h) dh \tag{1}$$

where $\mathbf{1}$ denotes the indicator function.

Note that this is simply negative log of the probability that a hypothesis sampled from $p_h$ achieves an error rate $\leq \varepsilon$ on a test set. $p_h$ may be any distribution over the hypothesis space; Boopathy et al. (2023) sets $p_h$ as a uniform distribution of models achieving a training set error $\leq \epsilon$. In practice, we may be interested in the case when $p_h$ is a distribution of models produced by an optimization process on a training set. This allows us to quantify the *additional* inductive bias required to generalize on top of any information provided by the training data. Critically, as Figure 1 illustrates, the specific choice of $p_h$ has a significant impact on the inductive bias. Intuitively, if the hypothesis distribution is more aligned with a task, fewer inductive biases are required to generalize.

Estimating this inductive bias by directly sampling hypotheses from a hypotheses space is computationally infeasible for large hypothesis spaces since the vast majority of hypotheses may not generalize well. Thus, we propose a two-phase approach to compute the inductive bias: first, we sample from the hypothesis space and compute an empirical distribution of test set error values $\mathbb{E}_{x \sim p_x}[L(h, x)]$. Few (or none) of these hypotheses may generalize at the desired error rate. Thus, we use the samples to model the test error distribution to estimate the probability of achieving test error $\leq \varepsilon$.

We also note that inductive bias in Equation 1 is a function of the desired error rate $\varepsilon$; it is not a function of a specific model or model class, although it is a function of the hypothesis space distribution $p_h$. However, we may use this definition to compute the inductive bias *provided* by a specific model under a specific hypothesis space by computing the amount of inductive bias required to generalize at the level of the model (i.e. by plugging in the model's test set error rate $\varepsilon$ into Equation 1). This allows us to understand how the inductive bias of a model is affected by the properties of the broader hypothesis space, and how it contributes to the model's generalization performance.

## 3.2 EFFICIENTLY SAMPLING FROM THE HYPOTHESIS SPACE

Here, we aim to efficiently sample hypotheses from $p_h$, where we assume $p_h$ includes only hypotheses fitted to a training set. We use two approaches: directly optimizing the parameters of a hypothesis (i.e. training a model on the training data), or a kernel-based sampling approach.

**Direct Optimization by Gradient Descent**  For hypothesis spaces with a known, finite-dimensional parameterization, it may be reasonable to set $p_h$ as a distribution of hypotheses produced by performing gradient descent on loss function $L$ evaluated on a training set of data $x$. For instance, $p_h$ may correspond to a distribution of neural networks after training from random initialization by gradient descent on a training set. Given $P$ parameters per hypothesis, performing each step of gradient descent takes $O(P)$ time, yielding $O(PT)$ time for $T$ optimization steps. Thus, producing $S$ samples requires $O(SPT)$ time.

**Kernel-based Sampling**  If the hypothesis space is very high-dimensional (or infinite-dimensional), it may be infeasible to directly optimize hypotheses by gradient descent. Instead, we formulate the problem of sampling from a hypothesis space as sampling from a Gaussian process, for which efficient algorithms have been extensively studied. We use an algorithm resembling the approach of Lin et al. (2023). The key principle behind our algorithm is to reparameterize the distribution of hypothesis output values on a test set in terms of a unit Gaussian. This allows us to easily and efficiently draw samples from this distribution in linear time (in terms of training set size).

We assume hypotheses $h$ are linearly parameterized with parameters $\theta \in \mathbb{R}^P$ as:

$$h(x) = \phi(x)\theta \tag{2}$$

where $\phi(x) \in \mathbb{R}^{k \times P}$ is a dimensional feature matrix and $k$ is the dimensionality of $h(x)$. Here, we set $p_h$ to include only the set of hypotheses that *interpolate* the training data. Given a set of $N$ training points $X$, their corresponding features $\phi(X) \in \mathbb{R}^{Nk \times P}$ and target model outputs $Y \in \mathbb{R}^{Nk}$, where $k$ represents output dimensionality, observe that if a hypothesis interpolates the training data, its parameters must satisfy:

$$Y = \phi(X)\theta \tag{3}$$

We may decompose $\theta$ into two terms:

$$\theta = \phi(X)^\dagger Y + \beta \tag{4}$$

where $\beta \in \mathbb{R}^P$ satisfies $\phi(X)\beta = 0$. The first term ensures the hypothesis fits the training data while the second term allows for variation between hypotheses. Finally, we set $\beta$ as a Gaussian with mean 0 and covariance $I - \phi(X)^\dagger \phi(X)$, where $\dagger$ represents pseudoinverse. This corresponds to setting the distribution of parameters $\theta$ as:

$$p_\theta(\theta) = \mathcal{N}(\phi(X)^\dagger Y, I - \phi(X)^\dagger \phi(X)) \tag{5}$$

This corresponds to a Gaussian process conditioned on the training points.

We aim to sample the value of $h$ on a test set $\bar{X}$ consisting of $n$ points. These values $h(\bar{X})$ may be computed as:

$$h(\bar{X}) = K(\bar{X}, X)\alpha^* + \sqrt{K(\bar{X}, \bar{X}) - K(\bar{X}, X)A^*} z \tag{6}$$

where $\alpha^*$ and $A^*$ are found as:

$$\alpha^* = \arg\min_\alpha ||Y - K(X, X)\alpha||_2^2 \tag{7}$$

$$A^* = \arg\min_A ||K(X, \bar{X}) - K(X, X)A||_F^2 \tag{8}$$

and $z$ is drawn from a unit Gaussian $\mathcal{N}(0, I)$. We find approximate solutions to these optimization problems by stochastic gradient descent. Pseudocode is provided in Algorithm 1, with additional details included in Appendix B.1. Producing $S$ samples requires a total of $O(nNk^2T + n^3k^3 + n^2Nk^3 + n^2k^2S)$ time where $T$ is the number of optimization steps.

---

**Algorithm 1** Kernel-based Sampling

---

1: **procedure** KERNELSAMPLING($X, Y, \bar{X}, n, k, T, S, \eta$)
2:     Initialize $\alpha$ and $A$ with zeros
3:     **for** $t = 1$ to $T$ **do**
4:         Randomly sample a mini-batch of training examples $(x, y) \in (X, Y)$
5:         Compute gradient $g_\alpha = 2K(X, x)(K(x, X)\alpha - y)$
6:         Compute gradient $g_A = 2K(X, x)(K(x, \bar{X})A - K(x, \bar{X}))$
7:         Update $\alpha$ using gradient descent: $\alpha \leftarrow \alpha - \eta g_\alpha$ (where $\eta$ is the learning rate)
8:         Update $A$ using gradient descent: $A \leftarrow A - \eta g_A$ (where $\eta$ is the learning rate)
9:     **end for**
10:    Compute mean $m = K(\bar{X}, X)\alpha$
11:    Compute square root of covariance $\sqrt{C} = \sqrt{K(\bar{X}, \bar{X}) - K(\bar{X}, X)A}$
12:    **Initialize** an empty list `samples`
13:    **for** $s = 1$ to $S$ **do**
14:        Sample $z \sim \mathcal{N}(0, I)$
15:        Compute sample $h(\bar{X}) = m + \sqrt{C}z$
16:        Append $h(\bar{x}_s)$ to `samples`
17:    **end for**
18:    **return** `samples`
19: **end procedure**

---

## 3.3 MODELING THE TEST ERROR DISTRIBUTION

Once we generate samples from the hypothesis space, we next aim to model the distribution of test set losses of sampled functions from the hypothesis space; this allows us to compute inductive bias.

To understand what the shape of the test loss distribution may be, we return to the assumptions in the kernel-based sampling of the hypothesis space. We assume that the regression targets $Y, \bar{Y}$ on the training and test sets respectively are constructed as $Y = \phi(X)\theta^*, \bar{Y} = \phi(\bar{X})\theta^*$ for some unknown parameters $\theta^*$.

The squared error between the prediction $h(\bar{X})$ and true value $\bar{Y}$ may be written as:

$$||h(\bar{X}) - \bar{Y}||_2^2$$
$$= ||\phi(\bar{X})\phi(X)^\dagger \phi(X)\theta^* + \phi(\bar{X})\beta - \phi(\bar{X})\theta^*||_2^2 \quad (9)$$

Figure 2: Fitting a scaled non-central Chi-squared distribution to an empirical distribution of mean squared errors of models drawn from a kernel-based Gaussian RBF hypothesis space on a restricted version of MNIST. Observe that the distribution closely models the empirical distribution.

We may write $\beta$ as $\beta = (I - \phi(X)^\dagger \phi(X))\xi$ where $\xi \in \mathbb{R}^P$ is distributed as a unit Gaussian. Then, the squared prediction error may be written as:

$$||h(\bar{X}) - \bar{Y}||_2^2 = ||\phi(\bar{X})(I - \phi(X)^\dagger \phi(X))(\xi - \theta^*)||_2^2 \quad (10)$$

Observe that this is a quadratic form of a Gaussian random variable $\xi - \theta^*$; thus, $||h(\bar{X}) - \bar{Y}||_2^2$ follows a generalized Chi-squared distribution.

In practice, to minimize the number of fit parameters when modeling the empirical error distribution, we fit the test loss using a scaled non-central Chi-squared (which is a special case of a generalized Chi-squared distribution); this has three fit parameters. These parameters are fit with maximum likelihood estimation. Figure 2 illustrates that this distribution is able to closely fit test errors of random hypotheses on a real dataset.

Since we model the test error distribution as a Chi-squared distribution, we need to approximate the negative log of the cumulative distribution function (CDF) given its parameters. Given a Chi-squared distribution with $k$ degrees of freedom and non-centrality parameter $\lambda$, we use the following approximation by Sankaran (Sankaran, 1959) for the CDF:

$$P(z; k, \lambda) \approx \Phi\left[ \frac{\left(\frac{z}{k+\lambda}\right)^h - (1 + hp(h - 1 - 0.5(2 - h)mp))}{h\sqrt{2p}(1 + 0.5mp)} \right], \quad (11)$$

where $\Phi$ is the CDF of a standard normal random variable and

$$h = 1 - \frac{2}{3}\frac{(k+\lambda)(k+3\lambda)}{(k+2\lambda)^2}, \qquad p = \frac{k+2\lambda}{(k+\lambda)^2}, \qquad m = (h-1)(1-3h).$$

Finally, we use a Chernoff bound-based approximation $\Phi(-z) \approx e^{\frac{-z^2}{2}}$ to finish the calculation. Specifically, since the inductive bias is given as the negative log probability of generalizing up to error rate $\varepsilon$ (see Equation 1), we simply compute the negative log of the approximated CDF after plugging in the desired $\varepsilon$ for $z$. Appendix B.3 provides further details on how the test error distribution is modeled.

### 3.4 BOUNDING THE APPROXIMATION ERROR

Next, we derive a bound on the approximation error of our estimate of required inductive bias. At a high level, the bound proceeds as follows: we first bound how closely our samples from the hypothesis distribution match the true distribution $p_h$, and then bound the error in our modeling of the test error distribution to arrive at a final bound on the amount of inductive bias.

**Theorem 1.** Suppose we are provided a hypothesis distribution $p_h$, input distribution $p_x$, loss function $L$ and desired error rate $\varepsilon$. Suppose we estimate $I(\varepsilon, p_h, p_x, L)$ by first sampling $n$ hypotheses $(h^1, h^2, ...h^n)$ iid from $q_h$ which is close to $p_h$ in the sense that

$$|\log p_h(h) - \log q_h(h)| \le \xi_h \quad (12)$$

for all $h$. We then compute the test losses of each hypothesis $\mathbb{E}_{x \sim p_x}[L(h^1, x)], \mathbb{E}_{x \sim p_x}[L(h^2, x)], \ldots \mathbb{E}_{x \sim p_x}[L(h^n, x)]$. Next, we model the distribution of test losses with a distribution $f(l; \alpha)$ where $\alpha$ represents a finite number of parameters. We assume $f$ has bounded support over $l$. We assume that knowing a finite number of moments of $f$ uniquely determines $\alpha$ in the sense that there exists $\tilde{f}$ such that $\tilde{f}(l; \mu) = f(l; \alpha)$ where $\mu$ represent $r$ moments of the distribution:

$$\mu = \int M(l) f(l; \alpha) dl \tag{13}$$

for some function $M(l)$. We assume $\log \tilde{f}$ is Lipschitz continuous with respect to $\mu$.

Denote the distribution of $\mathbb{E}_{x \sim p_x}[L(h, x)]$ when $h$ is drawn from $q_h$ as $q_l$. We assume $q_l$ can be closely modeled by $f(l; \alpha)$ in the following sense:

$$\max_l |\log q_l(l) - \log \tilde{f}(l; \bar{\mu})| \leq \xi_l \tag{14}$$

where $\bar{\mu} = \int M(l) q_l(l) dl$ are the moments of $q_l$. Given the empirical test loss distribution, we use the method of moments to estimate the parameters of $f$, yielding $\alpha^*$. Finally, suppose that the estimate of $I(\varepsilon, p_h, p_x, L)$ is computed as:

$$\tilde{I} = -\log \int_{-\infty}^{\varepsilon} f(l; \alpha^*) dl \tag{15}$$

Then with probability $1 - \sigma$, the approximation error of $\tilde{I}$ can be bounded as:

$$|\tilde{I} - I(\varepsilon, p_h, p_x, L)| \leq \xi_h + \xi_l + \frac{\kappa}{n} \sqrt{r \log \frac{2r}{\sigma}} \tag{16}$$

for a constant $\kappa$.

See Appendix A for a proof. Observe that the approximation error is bounded by three terms: the first corresponds to how accurately the hypothesis distribution can be sampled, the second corresponds to the modeling error of the test error distribution, and the third corresponds to the error from drawing a finite number of samples. Practically, the first term can be set to 0 is we are able to sample from $p_h$. Similarly, $\xi_l = 0$ is 0 if the test error distribution follows a scaled non-central Chi-squared distribution, which can be motivated theoretically and empirically as explained in Section 3.3. Thus, the remaining error is the finite sample approximation error which converges with rate $O(\frac{1}{n})$.

Note that instead of the method of moments, we use maximum likelihood estimation to estimate $\alpha$ since it is practically effective. Maximum likelihood estimation can also be shown to yield the same convergence rate asymptotically with $n$, although deriving a finite sample bound is more challenging.

## 4 EXPERIMENTAL RESULTS

We first evaluate inductive bias for various tasks under an infinite-dimensional kernel-based hypothesis space and interpret trends of inductive bias in terms of task properties. We then measure inductive bias in the context of a more restricted neural network hypothesis space.

### 4.1 INDUCTIVE BIAS INCREASES WITH TASK DIMENSIONALITY

We evaluate the inductive bias required to generalize on benchmark tasks across various domains: MNIST (Lecun et al., 1998), CIFAR-10 (Krizhevsky et al., 2009), 20-way 1-shot Omniglot (Lake et al., 2015) and inverted pendulum control (Florian, 2007). Classification tasks are treated as regression problems with a mean squared error loss function with one-hot-encoded labels. The hypothesis space is a kernel-based hypothesis space using a Gaussian RBF kernel constructed as $K(x_1, x_2) = e^{-\frac{1}{2}||x_1 - x_2||_2^2} I$. Note that the hypotheses from this space are constrained to fit the training data; thus, our inductive bias measure quantifies additional information required to generalize on top of the training data. We use Algorithm 1 to sample hypotheses from this space and evaluate their mean squared error on the test set of each task. Note that this hypothesis space is infinite-dimensional; directly optimizing in the space is not feasible. Due to computational constraints,

Table 1: Gaussian RBF model class results, compared with bounds from Boopathy et al. (2023) for image classification datasets (MNIST, CIFAR-10), Omniglot, and Inverted Pendulum tasks. *Our version of the Inverted Pendulum task differs somewhat from Boopathy et al. (2023).

| Task | Upper Bound (Boopathy et al., 2023) | Our Results |
|---|---|---|
| Inverted Pendulum | $4.41 \times 10^9$ bits* | 29 bits |
| MNIST | $1.48 \times 10^{16}$ bits | 2568 bits |
| CIFAR-10 | $3.43 \times 10^{32}$ bits | 2670 bits |
| Omniglot | $1.79 \times 10^{145}$ bits | 2857 bits |

gradient descent in Algorithm 1 is not run until full convergence; thus, sampled hypotheses may not interpolate the training data. Nevertheless, we find that the corresponding distribution of losses stabilizes after a small number of epochs (see Appendix B.1 Figure 5). Once we have a distribution of mean-squared errors, we use the approach of Section 3.3 to compute inductive bias. Appendix B includes further experimental details.

Figure 3 plots the fitted Chi-squared distributions to the test set errors for the four tasks. Table 1 showcases the resulting inductive bias estimates and compares them to Boopathy et al. (2023)'s prior upper bounds on inductive bias; Boopathy et al. (2023) use a similarly high-dimensional hypothesis space as our kernel-based hypothesis space. Note that given our choice of model class, our measure is *many orders of magnitude* lower than Boopathy et al. (2023)'s prior upper bound. This is because Boopathy et al. (2023) computed upper bounds of inductive bias, while we used a more precise estimation method. Further, our hypothesis space, although quite broad, is different than theirs and potentially more restricted, leading to fewer bits being needed to narrow down the well-generalizing hypotheses. The particular number extracted for each task also depends on the particular cutoff test set error rate $\varepsilon$ chosen; we chose $\leq 0.001$

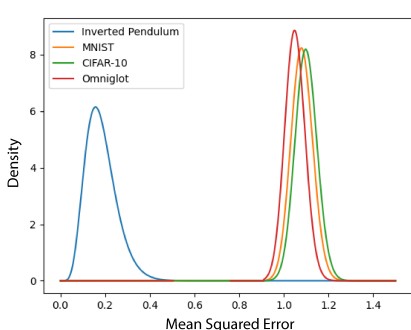

Figure 3: Fitted scaled non-central Chi-squared distributions for the test set errors on MNIST, CIFAR-10, Omniglot, and Inverted Pendulum tasks.

as the required quadratic loss for all four tasks as a measure of being well-generalizing. This roughly corresponds to the performance of typical competitive models on each of the tasks.

In general, we see that tasks with more intrinsic dimensionality require more bits of inductive bias to generalize well; in particular, Inverted Pendulum < MNIST < CIFAR-10 < Omniglot, which matches our expectations from Boopathy et al. (2023)'s results. Interestingly, if we examine the main peaks in the Chi-squared distribution from Figure 3, the Omniglot task errors appear shifted further to the left which would suggest a lower inductive bias. However, Omniglot has a lower variance in hypotheses that interpolate the training data, leading to a lower probability that the error is very small, compared to MNIST or CIFAR-10.

## 4.2 RESTRICTING THE BASE HYPOTHESIS SPACE REDUCES REQUIRED INDUCTIVE BIAS

Next, we measure the inductive bias of various models trained on MNIST under a more restricted, but still expressive hypothesis space. Specifically, we consider the hypothesis space expressible by a high-capacity ReLU-activated fully-connected neural network with 9 layers and 512 units per hidden layer. Appendix B describes the details of how the hypothesis space is constructed. Note that our inductive bias measure is a function of the desired test set error rate ($\varepsilon$ in Equation 1); previously, we set the desired error rate as a fixed value for each task. However, we may also compute the inductive bias *provided* by different models. Following Boopathy et al. (2023), we compare models by evaluating the test set error of the models and plugging this error into $\varepsilon$ in Equation 1. In other words, the inductive bias provided by a model is the amount of information required to achieve the error rate of the model. We evaluate four different models: a linear model,

a decision tree, a wide and shallow neural network (depth 3, widths 512 and 256), and a deep and narrow neural network (depth 6, width 64). See Appendix B for additional model details.

In Figure 4, we find that all models provide very little inductive bias (in contrast to the much larger numbers observed for the kernel hypothesis space earlier); the largest measured inductive bias is below 0.01 bits. On this scale, even though all models achieve similar errors, the relative difference between the inductive bias of different models is quite large. This reverses the trend observed in the high-dimensional hypothesis space of Boopathy et al. (2023) where different models with quite different error rates contained similar amounts of inductive bias. In other words, within a more restricted hypothesis space, the relative difference between the inductive biases of different models is accentuated. The results demonstrate that within the neural network hypothesis space, the *additional* inductive bias of practical models may be quite small. That is, neural networks themselves provide enormous inductive bias.

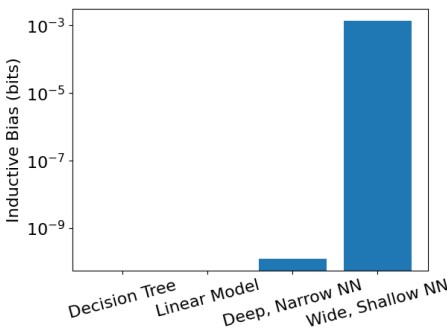

Figure 4: Inductive bias of different models trained on MNIST under a neural network hypothesis space. The decision tree and linear model have inductive bias smaller than machine precision and are thus estimated as 0.

## 5 DISCUSSION

Our results reveal that different tasks require different levels of inductive bias, with higher dimensional tasks demanding greater amounts. In particular, with expressive kernel-based hypothesis spaces, the required inductive bias can be higher for high-dimensional tasks such a Omniglot compared to lower-dimensional tasks such as CIFAR-10 *even when the lower-dimensional task may be intuitively simpler*. This curse of dimensionality occurs due to an exponential explosion of the size of the hypothesis space with the task dimension: intuitively, each additional dimension of variation in a task increases the *dimensionality* of the hypothesis space by a constant factor. Our findings confirm previous research and highlight the importance of the choice of model class, particularly for high-dimensional problems.

We also find that neural networks as a model class, inherently encode massive amounts of inductive bias. The choice of neural networks themselves provides a much greater inductive bias than specific architectural choices, although our measure also reveals that architectural choices can provide significant inductive bias. This observation suggests that the strong smoothness (Li et al., 2018) and compositionality (Mhaskar et al., 2017) constraints of neural networks align well with the properties of realistic tasks. Consequently, these models naturally embody the inductive bias required for a wide range of tasks, underscoring their prevalence and success across various domains.

We note that our empirical results are restricted to two specific choices of hypothesis spaces: a Gaussian RBF kernel-based hypothesis space and a fixed neural network hypothesis space. However, our approach is applicable to *general* hypothesis spaces. Future work may be able to extend our inductive bias quantification to other hypothesis space choices.

We propose two potential ways of using our inductive bias quantification. First, it provides an information-theoretic interpretation of the advantages of particular model architectures for specific tasks. By quantifying the amount of inductive bias associated with different architectural choices, researchers can gain insights into how specific design decisions affect the model's ability to generalize. This understanding helps identify which architectural features contribute most significantly to improved performance and informs the development of more tailored and task-specific models.

Second, the inductive bias measure serves as a quantitative guide for developing tasks that require greater inductive bias. By precisely estimating the amount of inductive bias needed for a given task, researchers can intentionally design challenging benchmarks that push the boundaries of machine learning capabilities. We hope this can encourage the development of more powerful model architectures and learning algorithms that drive the field forward.

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

# A PROOF OF THEOREM 1

*Proof.* Denote the probability distribution over $\mathbb{E}_{x \sim p_x}[L(h, x)]$ when $h$ is drawn from $p_h$ as $p_l(l)$. From the definition of $\tilde{I}$ and $I(\varepsilon, p_h, p_x, L)$, it is known that:

$$|\tilde{I} - I(\varepsilon, p_h, p_x, L)| = |-\log \int_{-\infty}^{\varepsilon} f(l; \alpha^*) dl + \log \int_{-\infty}^{\varepsilon} p_l(l) dl| \tag{17}$$

Expressing the difference of logs as a log of a ratio:

$$|\tilde{I} - I(\varepsilon, p_h, p_x, L)| = \left| -\log \frac{\int_{-\infty}^{\varepsilon} f(l; \alpha^*) dl}{\int_{-\infty}^{\varepsilon} p_l(l) dl} \right| \tag{18}$$

Observe that this expression can be upper bounded by:

$$\max_l |\log f(l; \alpha^*) - \log p_l(l)| \tag{19}$$

To see this, denote $k = \max_l |\log f(l; \alpha^*) - \log p_l(l)|$. Then,

$$f(l; \alpha^*) \leq e^k p_l(l) \tag{20}$$

This implies:

$$\log \frac{\int_{-\infty}^{\varepsilon} f(l; \alpha^*) dl}{\int_{-\infty}^{\varepsilon} p_l(l) dl} \leq \log \frac{\int_{-\infty}^{\varepsilon} e^k p_l(l) dl}{\int_{-\infty}^{\varepsilon} p_l(l) dl} = \log e^k \frac{\int_{-\infty}^{\varepsilon} p_l(l) dl}{\int_{-\infty}^{\varepsilon} p_l(l) dl} = k \tag{21}$$

Similarly, $-\log \frac{\int_{-\infty}^{\varepsilon} f(l; \alpha^*) dl}{\int_{-\infty}^{\varepsilon} p_l(l) dl}$ can be upper bounded by $k$. Thus,

$$|\tilde{I} - I(\varepsilon, p_h, p_x, L)| \leq \max_l |\log f(l; \alpha^*) - \log p_l(l)| \tag{22}$$

We can upper bound the absolute value as:

$$\max_l |\log f(l; \alpha^*) - \log p_l(l)| \leq \max_l |\log f(l; \alpha^*) - \log q_l(l)| + \max_l |\log q_l(l) - \log p_l(l)| \tag{23}$$

We first bound the first term. Reparameterizing $f$ in terms of moments $\mu$:

$$\max_l |\log f(l; \alpha^*) - \log q_l(l)| = \max_l |\log \tilde{f}(l; \mu^*) - \log q_l(l)| \tag{24}$$

where $\mu^*$ are moment estimates computed as sample averages. We upper bound the absolute value as:

$$\max_l |\log \tilde{f}(l; \mu^*) - \log q_l(l)| \leq \max_l |\log \tilde{f}(l; \mu^*) - \log \tilde{f}(l; \bar{\mu})| + \max_l |\log \tilde{f}(l; \bar{\mu}) - \log q_l(l)| \tag{25}$$

Note that $\max_l |\log \tilde{f}(l; \bar{\mu}) - \log q_l(l)|$ is bounded by $\xi_l$. Since $f$ has bounded support, by Hoeffding's inequality, the deviation from the mean of a single element of $\mu^*$ can be bounded as:

$$P(|\mu_i^* - \bar{\mu}_i|^2 \geq \frac{t}{r}) \leq 2e^{-C\frac{n^2 t}{r}} \tag{26}$$

for some constant $C$. Using the union bound:

$$P(||\mu_i^* - \bar{\mu}_i||^2 \geq t) \leq 2re^{-C\frac{n^2 t}{r}} \tag{27}$$

We set $\sigma = 2re^{-C\frac{n^2 t}{r}}$, which yields:

$$t = \frac{r}{Cn^2} \log \frac{2r}{\sigma} \tag{28}$$

Thus, with probability $1 - \sigma$:

$$P(||\mu_i^* - \bar{\mu}_i|| \leq \frac{1}{n}\sqrt{\frac{r}{C} \log \frac{2r}{\sigma}}) \tag{29}$$

Since $\log \tilde{f}$ is Lipschitz continuous with respect to $\mu$, with probability $1 - \sigma$:

$$\max_l |\log \tilde{f}(l; \mu^*) - \log \tilde{f}(l; \bar{\mu})| \leq \frac{\kappa}{n}\sqrt{r \log \frac{2r}{\sigma}} \tag{30}$$

for some constant $\kappa$. Thus, we may bound $\max_l |\log f(l; \alpha^*) - \log q_l(l)|$ as:

$$\max_l |\log f(l; \alpha^*) - \log q_l(l)| \leq \xi_l + \frac{\kappa}{n}\sqrt{r \log \frac{2r}{\sigma}} \tag{31}$$

Next, we bound $\max_l |\log q_l(l) - \log p_l(l)|$ using the bound on $\max_h |\log q_h(h) - \log p_h(h)|$. Note that

$$\log q_l(l) - \log p_l(l) = \log \frac{\int_{h:\mathbb{E}_{x \sim p_x}[L(h,x)]=l} q_h(h)dh}{\int_{h:\mathbb{E}_{x \sim p_x}[L(h,x)]=l} p_h(h)dh} \leq \log \frac{\int_{h:\mathbb{E}_{x \sim p_x}[L(h,x)]=l} e^{\xi_h} p_h(h)dh}{\int_{h:\mathbb{E}_{x \sim p_x}[L(h,x)]=l} p_h(h)dh} = \xi_h \tag{32}$$

Similarly,

$$\log p_l(l) - \log q_l(l) = \log \frac{\int_{h:\mathbb{E}_{x \sim p_x}[L(h,x)]=l} p_h(h)dh}{\int_{h:\mathbb{E}_{x \sim p_x}[L(h,x)]=l} q_h(h)dh} \leq \log \frac{\int_{h:\mathbb{E}_{x \sim p_x}[L(h,x)]=l} e^{\xi_h} q_h(h)dh}{\int_{h:\mathbb{E}_{x \sim p_x}[L(h,x)]=l} q_h(h)dh} = \xi_h \tag{33}$$

Therefore,

$$\max_l |\log q_l(l) - \log p_l(l)| \leq \xi_h \tag{34}$$

Combining all the inequalities, with probability $1 - \sigma$:

$$|\tilde{I} - I(\varepsilon, p_h, p_x, L)| \leq \xi_h + \xi_l + \frac{\kappa}{n}\sqrt{r \log \frac{2r}{\sigma}} \tag{35}$$

$\square$

# B  ADDITIONAL EXPERIMENTAL DETAILS

## B.1  KERNEL-BASED SAMPLING DETAILS

Using the decomposition of $\theta$, $h(\bar{X}) \in \mathbb{R}^{nk}$ can be expressed as:

$$h(\bar{X}) = \phi(\bar{X})\phi(X)^\dagger Y + \phi(\bar{X})\beta \tag{36}$$

where $\phi(\bar{X}) \in \mathbb{R}^{nk \times P}$. Note that this is distributed as a Gaussian with mean $\phi(\bar{X})\phi(X)^\dagger Y$ and covariance matrix $\phi(\bar{X})(I - \phi(X)^\dagger\phi(X))\phi(\bar{X})^T = \phi(\bar{X})\phi(\bar{X})^T - \phi(\bar{X})\phi(X)^\dagger\phi(X)\phi(\bar{X})^T$. We may express these quantities in terms of the *kernel* corresponding to features $\phi(x)$. The kernel is defined a $k \times k$ matrix:

$$K(x_1, x_2) = \phi(x_1)\phi(x_2)^T \tag{37}$$

In our experiments, we use a Gaussian radial basis function (RBF) kernel. We denote $K(X, X) \in \mathbb{R}^{Nk,Nk}$ and $K(\bar{X}, X) \in \mathbb{R}^{nK,Nk}$ as the kernels between all pairs of training points and pairs of test and training points respectively. Then, assuming $N < P$, $h(\bar{X})$ has mean $K(\bar{X}, X)K(X, X)^{-1}Y$ and covariance $K(\bar{X}, \bar{X}) - K(\bar{X}, X)K(X, X)^{-1}K(X, \bar{X})$. Thus, we may express $h(\bar{X})$ as:

$$h(\bar{X}) = K(\bar{X}, X)K(X, X)^{-1}Y + \sqrt{K(\bar{X}, \bar{X}) - K(\bar{X}, X)K(X, X)^{-1}K(X, \bar{X})}z \tag{38}$$

where $z \in \mathbb{R}^{nk}$ is distributed as a unit Gaussian and $\sqrt{}$ denotes matrix square root. Computing this quantity directly for a given choice of $z$ may be computationally challenging since it requires storing and inverting the kernel matrix $K(X, X)$. Thus, we instead approximate solutions to $K(X, X)^{-1}Y$ and $K(X, X)^{-1}K(X, \bar{X})$ via gradient descent. Specifically, we define $\alpha^* = K(X, X)^{-1}Y$ and $A^* = K(X, X)^{-1}K(X, \bar{X})$. Note that these can be found as solutions to the following optimization problems:

$$\alpha^* = \arg\min_\alpha ||Y - K(X, X)\alpha||_2^2 \tag{39}$$

$$A^* = \arg\min_A ||K(X, \bar{X}) - K(X, X)A||_F^2 \tag{40}$$

We find approximate solutions to these optimization problems by stochastic gradient descent. Once $\alpha^*$ and $A^*$ are found, samples of $h(\bar{X})$ may be computed as:

$$h(\bar{X}) = K(\bar{X}, X)\alpha^* + \sqrt{K(\bar{X}, \bar{X}) - K(\bar{X}, X)A^*}z \tag{41}$$

Assuming a constant time kernel computation per element, performing stochastic gradient descent on $\alpha$ and $A$ for one iteration requires $O(Nk)$ and $O(nNk^2)$ time respectively; thus, for $T$ optimization steps, the time complexity is $O(nNk^2T)$. Once computed, sampling $h(\bar{X})$ requires $O(n^3k^3 + n^2Nk^3)$ time to determine the constants in the above equation, and an additional $O(n^2k^2)$ per sample. Thus, producing $S$ samples requires a total of $O(nNk^2T + n^3k^3 + n^2Nk^3 + n^2k^2S)$ time. Importantly, this time is *linear* in the training set size and *constant* in the parameter size; this efficiency is critical for large datasets and high-dimensional hypothesis spaces.

We use a Gaussian RBF kernel constructed as $K(x_1, x_2) = e^{-\frac{1}{2}||x_1-x_2||_2^2}I$. Due to the computational cost of computing the full gradients $g_\alpha$ and $g_A$ (both of which have sizes scaling linearly with the training set size), we compute the gradient in steps. Specifically, for $g_\alpha$, $K(x, X)\alpha$ is computed by splitting the training set into groups $X_i$ (and correspondingly $\alpha$ into $\alpha_i$) and summing the contribution of each $K(x, X_i)\alpha_i$. An analogous grouping is done to compute $g_A$.

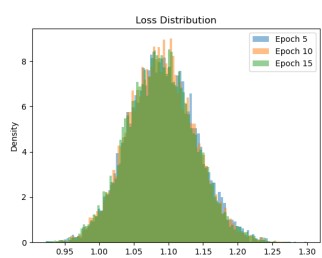

Figure 5: Distribution of hypothesis losses for MNIST after 5, 10, and 15 epochs of gradient descent. Notice that the update in distribution is minimal.

For experiments on MNIST, we use a learning rate of $0.0001$ to optimize $\alpha$ and a learning rate of $0.00001$ to optimize $A$. We use a batch size of $128$ and a group size of $2048$. Training is performed for $20$ epochs. See Figure 5 for an analysis of the convergence of the resulting loss distribution.

For experiments on CIFAR-10, we use a learning rate of $0.001$ to optimize $\alpha$ and a learning rate of $0.0001$ to optimize $A$. We use a batch size of $64$ and a group size of $1024$. Training is performed for $20$ epochs.

For experiments on Omniglot, we use a learning rate of $0.0001$ to optimize $\alpha$ and a learning rate of $0.00001$ to optimize $A$. We use a batch size of $10$ and a group size of $100$. Training is performed for $1$ epoch.

For experiments on the inverted pendulum task, we use a learning rate of $0.001$ to optimize $\alpha$ and a learning rate of $0.0001$ to optimize $A$. We use a batch size of $64$ and a group size of $1024$. Training is performed for $500$ epochs.

### B.2 NEURAL NETWORK HYPOTHESIS SPACE DETAILS

The architecture of our base hypothesis space is constructed as follows: the input is linearly projected to a $512$ dimensional vector, followed by $9$ more fully connected layers of dimensionality $512$. Finally, the output is linearly projected to a 10-dimensional output to predict the one-hot encoded label. All layers include a bias term. Each fully connected layer except the final one is followed by a ReLU non-linearity. No additional components such as normalization are used.

The model is trained on a mean squared error loss using Adam with a learning rate of $0.001$. Training is conducted for $10$ epochs with a batch size of $128$. We sample $100$ hypotheses from this hypothesis space by using different random initializations and orderings of training points during training.

Within the neural network hypothesis space, we consider the performance of four models: decision tree, linear model, deep and narrow neural network and a wide and shallow neural network. All models are trained to minimize mean squared error training loss.

For the decision tree model, Gini impurity is used to evaluate split quality. At each node, the best split is chosen. A minimum of $2$ samples are required to split a sample. Predictions are clipped to be in the range $[0, 1]$ before computing mean-squared error with one-hot encoded labels.

For the linear model, ordinary least squares fitting is used to fit the model. Predictions are clipped to be in the range $[0, 1]$ before computing mean-squared error with one-hot encoded labels.

For the deep and narrow neural network, we construct a neural network consisting of $6$ fully connected layers each of output $64$ except the final output which has dimensionality $10$. Each layer except the

last one is followed by a ReLU non-linearity. No additional components such as normalization are used. The model is trained on a mean squared error loss using Adam with a learning rate of $0.001$.

For the wide and shallow neural network, we construct a neural network consisting of a fully connected layer with output dimensionality $512$ followed by a fully connected layer with output dimensionality $256$ followed by a final fully connected layer with output dimensionality $10$. Each layer except the last one is followed by a ReLU non-linearity. No additional components such as normalization are used. The model is trained on a mean squared error loss using Adam with a learning rate of $0.001$.

### B.3 DISTRIBUTION FITTING DETAILS

We sample $100000$ hypotheses from the hypothesis space and compute the test set error for each one. Given the empirical distribution of test set errors, we fit a three-parameter scaled non-central Chi-squared distribution to match the data. We use maximum likelihood estimation to determine the fit parameters. Once the distribution parameters are determined, we use approximations to quantify the log of the cumulative distribution function as described in the main text.

### B.4 TASKS

For all tasks, desired error rates are set as the values provided in Boopathy et al. (2023) unless otherwise specified.

**MNIST & CIFAR-10**  We use the base MNIST and CIFAR-10 datasets without modification.

**ImageNet**  We use standard ImageNet normalization and random cropping.

**Omniglot**  We consider 20-way 1-shot Omniglot classification. In this setting, each input consists of 20 images, 1 from each of 20 alphabets, and the goal is to predict the class of a new image from one of the 20 seen alphabets. We encode this task in the following form: the inputs $x$ consist of 21 images, the first 20 of which correspond to training images and the last one of which corresponds to the evaluation image. The input is flattened to remove all spatial structure. The desired output is a one-hot encoded 20-dimensional vector of which of the 20 training images matches the class of the evaluation image. We generate a training set of size 1000 and a test set of size 100, drawn from the Omniglot background and evaluation alphabets respectively.

**Inverted Pendulum Task**  We consider the following inverted pendulum control task: an inverted pendulum with angle $\theta \in \mathbb{R}$ and angular velocity $\omega \in \mathbb{R}$ has the following dynamics:

$$\dot{\theta} = \omega \tag{42}$$

$$\dot{\omega} = \sin \theta + u \tag{43}$$

where $u \in \mathbb{R}$ is a control action. The goal is to minimize the time average of the following cost:

$$C(u, \theta, \omega) = \frac{1}{2}u^2 + 24\theta^2 + (8\theta + 4\omega)(\theta - \sin \theta) \tag{44}$$

The first term encourages small control actions. The second term encourages the inverted pendulum to remain at rest at $\theta = 0$. The third term is added to allow an analytically tractable optimal control. Intuitively, it penalizes when future values of $\theta$ (as represented by $8\theta + 4\omega$) are far from $0$ (as represented by $\theta - \sin \theta$). Note that this term is on the order of $O(\theta^3)$; for small $\theta$, the second term dominates.

The optimal cost to go, or value function (i.e. the optimal total cost over all future time steps) given current state $\theta, \omega$ is:

$$V(\theta, \omega) = 14\theta^2 + 8\theta\omega + 2\omega^2 \tag{45}$$

To verify this, note that the optimal cost obeys the Bellman equation:

$$\min_u \frac{\partial}{\partial \theta}V(\theta, \omega)\dot{\theta} + \frac{\partial}{\partial \omega}V(\theta, \omega)\dot{\omega} + C(u, \theta, \omega) = 0 \tag{46}$$

Substituting in the expressions from above:

$$\min_u[(28\theta + 8\omega)\omega + (8\theta + 4\omega)(\sin\theta + u) + \frac{1}{2}u^2 + 24\theta^2 + (8\theta + 4\omega)(\theta - \sin\theta)] = 0 \quad (47)$$

Setting the derivative with respect to $u$ to 0, the optimal $u$ must satisfy:

$$8\theta + 4\omega + u = 0 \quad (48)$$

Thus, $u = -8\theta - 4\omega$. Plugging this back into the Bellman equation:

$$(28\theta + 8\omega)\omega + (8\theta + 4\omega)(\sin\theta - 8\theta - 4\omega) + \frac{1}{2}(-8\theta - 4\omega)^2 + 24\theta^2 + (8\theta + 4\omega)(\theta - \sin\theta) = 0 \quad (49)$$

Observe that all terms on the left-hand side cancel; thus $V(\theta, w)$i is the correct value function for this optimal control problem. Given $\theta, \omega$, the optimal control action is:

$$u = -8\theta - 4\omega \quad (50)$$

In this task, inputs $(\theta, \omega)$ are drawn from a uniform distribution over $[-\pi, \pi] \times [-1, 1]$. Desired outputs $u$ are constructed as above. We generate a training set of size 10000 and a test set of size 100.

## B.5 COMPUTING INFRASTRUCTURE

Experiments are run on a computing cluster with GPUs ranging in memory size from 11 GB to 80 GB.

