# OpenReview forum: "Towards Exact Computation of Inductive Bias"
_ICLR.cc/2024/Conference — Submitted to ICLR 2024_

### Official Review · Reviewer_VSJh · 2023-10-28

**Soundness:** 3 good
**Presentation:** 2 fair
**Contribution:** 2 fair
**Rating:** 5
**Confidence:** 3

**Summary:**

The paper investigates the relevant problem of quantifying the level of inductive bias needed to specify well-generalizing models within a hypothesis of models.

After introducing the definitions of inductive biases, hypothesis space and model classes, an explicit formula to measure the amount of inductive bias is provided. This formula depends on a pre-specified error rate and measures how much inductive bias is required to achieve a certain test error given the hypothesis space under consideration. This quantity can be estimated by first sampling from the hypothesis space (which translates to sampling models that attain a low training error) and subsequently estimating the distribution of test error associated with the previously sampled models.

The paper focuses on two types of hypothesis spaces: 1) neural networks trained by gradient descent (each element of such a space corresponding to a different initialisation of the weights) and 2) Gaussian Processes with RBF kernel.

The experiments, performed on MNIST, CIFAR-10, Omniglot and inverted pendulum control, show that the proposed approach provides estimates of the amount of inductive bias that are in line with previous works (despite the latter being based on upper bounds).

**Strengths:**

The explicit quantification and characterisation of inductive biases is a very relevant problem in machine learning. This paper introduces a new method (to the best of the reviewer's knowledge) to measure the amount of inductive bias necessary to achieve a pre-specified level of test error for a given task. In principle I find the proposed method described in Eq. 1 quite clear and sound.

The paper is generally well written and the related works and necessary background concepts are carefully introduced in Section 2 and 3 respectively.

As mentioned by the authors, the proposed approach stands in contrast to previous works which were mainly based on upper bounds  and were limited to specific hypothesis spaces.

**Weaknesses:**

While I find the paper interesting and well-motivated, I believe its present form contains a number of weaknesses that limit its value:

- Some parts of the paper require further analysis and should be further clarified. In particular, I found it quite hard to understand Section 4.2 which explores the proposed method in the context of the hypothesis space entailed by neural networks. I believe this part should be explained more carefully and more details about the analysis should be provided. Appendix B mainly provided details about the model architecture and hyperparameters but not on how, for example, Fig 4 is obtained.

- As explicitly mentioned by the authors, inductive biases can arise at several levels, including but not limited to, the choice of the architecture and optimisers. Such aspects are not investigated as the reported experiments only focus on MLPs of different width and depth. It is not clear if the proposed method could be applied in this setting. More empirical evidence should be provided.

-  Equation 1 critically depends on the value of \epsilon which, in practice (see section 4.1) is arbitrarily chosen. I understand that in section 4.1 this is enough to demonstrate the value of the method, but in more general settings choosing an appropriate value of \epsilon may be less obvious.

- the derivation in Section 3.3 is based on kernel hypothesis space and it is not clear to what extent it transfers to different hypothesis spaces.

- The experiments contain quite simple tasks and architectures. It would be interesting to explore the feasibility of the proposed approach in more challenging settings (e.g. more complex architectures like CNNs or Transformers)

**Questions:**

See weaknesses section.

---

> ### Author Response · Authors · 2023-11-21
>
> Thank you for your valuable comments and constructive feedback.
>
> **Clarity**
>
> Thank you for your suggestions on how to improve the clarity of our work. We will carefully clarify these points in our revision.
>
> **Experimental results in additional settings**
>
> Thank you for your suggestions. We believe evaluating the inductive biases arising from different architectures and optimizers as well as over more tasks is an important direction for future work.
>
> **Choice of $\epsilon$**
>
> Indeed, as the reviewer notes, an appropriate choice of $\epsilon$ may not be obvious in all settings. However, as a general guideline, we propose choosing $\epsilon$ to correspond to competitive or state-of-the-art performance on a task. This allows the measure of inductive bias to quantify how much inductive bias is required to achieve what may be considered "good" performance on the task.
>
> **Test error distribution model**
>
> Indeed, as the reviewer notes, the derivation of the test error distribution in Section 3.3 relies on the kernel-based hypothesis space. We regard how to extend this analysis to more general hypothesis spaces as an important direction for future work.

---

### Official Review · Reviewer_9oUe · 2023-10-29

**Soundness:** 2 fair
**Presentation:** 3 good
**Contribution:** 2 fair
**Rating:** 3
**Confidence:** 3

**Summary:**

The paper studies the problem of quantitatively estimating the inductive bias required to achieve a desired level of test generalization in regression tasks. The estimator is based on the negative log probability of achieving such level, which can be computed by first sampling different hypotheses, by training them using the training data (in the case of kernel methods, the solution can be derived in an analytic form and in the case of standard neural networks, the solution is obtained through standard gradient-based optimization), by fitting a specific distribution (chi-squared) on the test mean squared errors achieved by the sampled hypotheses and finally by determining the probability of achieving a test error lower than the desired one. The estimator is compared against the recent one proposed in [1] across several datasets such as the inverted pendulum, MNIST, CIFAR-10 and Omniglot, thus highlighting improve tightness to the real value.

[1] Model-Agnostic Measure of Generalization Difficulty. ICML 2023

**Strengths:**

1. The paper is clear and well-written (**Clarity**)
2. The considered problem of estimating inductive bias is relevant and worth of being studied (**Significance**)

**Weaknesses:**

1. Several theoretical results are overstated and it is not clear what is their novelty compared to existing ones. For instance, the result about the test error distribution (Section 3.3) follows directly from a known one, i.e. it is well known that the sum of squared errors for a linear regressor follows a chi-squared distribution.  Additionally, the statistical result about the finite sample approximation (Section 3.4) is already known for a chi-squared distribution. Why not simply casting the discussion in such terms? (**Originality**)
2. It is not clear why it is important (i) to introduce new terminology and to distinguish between hypothesis space and model class, (ii) to consider the first space as a superset of the second one and (iii) to define the inductive bias as the negative log probability of sampling a hypothesis in the model class. Regarding the first point and as far as I understand, the inductive bias is merely defined by choosing the proper hypothesis space (architecture and hyperparameters). Regarding the second point, the provided definition of hypothesis space and model class is not consistent with the experiments. Indeed, in section 4.2, the hypothesis space consists of neural networks, whereas the model class includes decision trees, linear regressors as well as neural networks. Clearly, the model class is not a subset of the hypothesis space, as mentioned in Section 3.1. Regarding the third point, especially for experiments about kernel regression, it is not clear why the proposed prior distribution on the hypothesis space represents a meaningful inductive bias (i.e. once you have chosen a kernel and a bandwidth in the case of a Gaussian kernel, there are no additional biases to introduce) and therefore what is the proposed quantity computing? (**Quality/Significance**)
3. The experimental analysis is quite limited and could be more thourough. This could help to deepen the insights drawn from the analysis. For instance, by comparing different inductive biases inside each function class (e.g. different kernels for kernel regressors, different architectures for neural networks, or different choices of hyperparameters). (**Quality/Significance**)
4. Code is not available. Therefore, results are not reproducible (**Reproducibility**)

**Questions:**

Can you please elaborate more on the 4 above-mentioned weaknesses?

Moreover, it is not clear to me which insights can be drawn from Figure 3 and from the quantitive values provided in Table 1. Indeed, the conclusions drawn from these experiments might not be realistic, even misleading, as the models might be underfitting the data (see MNIST, CIFAR-10 and Omniglot). Perhaps, can you show the training losses to clarify this?

---

> ### Author Response · Authors · 2023-11-21
>
> Thank you for your valuable comments and constructive feedback.
>
> **Relationship to prior theoretical results**
>
> Indeed, as the reviewer notes, our proposed method is built on known results such as the approximation error of chi-squared distribution estimation. Our contribution is to use these tools to more precisely estimate inductive bias and provide bounds on the approximation error of our estimate.
>
> We would be happy to clarify this in our revision.
>
> **Concerns on definitions and terminology**
>
> Regarding the terminology and definitions related to inductive bias, we highlight that these are not conventions we introduce; rather, they have been proposed in prior literature (see Chollet (2019) and Boopathy et al. (2023)). We adopt these definitions to be most clear.  In particular, a hypothesis space corresponds to the space of all functions relevant to consider on a task. A model class is the space of hypotheses used to generalize on a task.
>
> **Neural network hypothesis space**
>
> In Section 4.2, we suppose the hypothesis space of large neural nets is sufficiently expressive to express decision trees and linear regressors. In the case of linear regressors in particular, it is possible to explicitly set the parameters of a large neural network to express any linear function.
>
> **Concerns on kernel-based hypothesis space**
>
> In our experiments, we measure how much inductive bias is required to generalize on a task in the context of a kernel hypothesis space. For example, we may quantify the inductive bias of a neural network model class in the context of a kernel hypothesis space. We believe that kernel-based hypothesis space is a natural choice for a hypothesis space since, in general, any function has non-zero probability density under a kernel-based hypothesis space. Thus, it is broad enough to capture any model class, while simultaneously allowing for bias toward the most reasonable hypotheses (e.g. smooth, compositional functions) depending on the choice of kernel.
>
> **Concerns about experimental analysis**
>
> Thank you for your suggestions on how to improve our experimental analysis. We believe further evaluating different architecture, kernel, and hyperparameter choices is an important direction for future work.

---

### Official Review · Reviewer_Fhpo · 2023-11-05

**Soundness:** 1 poor
**Presentation:** 1 poor
**Contribution:** 2 fair
**Rating:** 3
**Confidence:** 2

**Summary:**

The paper proposes and argues for a mathematical definition of inductive bias that can be computed for hypothesis spaces of models based on the fraction of well-generalizing models. The computation is then enabled for kernel-based sampling. Experiments are conducted on image classification datasets and in comparison to a prior method that upper bounds inductive bias.

**Strengths:**

- defining and quantifying inductive bias are both important problems to tackle
- Figure 1 nicely illustrates the idea of the definition
- Definition of inductive bias clear (Definition 1) but novelty unclear

**Weaknesses:**

Overall, I feel unable to properly review the paper because it seems to be in an early draft stage where the experiments are not entirely finished, the method only partially developed, and the related work is not completely clear, yet. I can see that there might be interesting ideas in the paper but in its current form, this paper seems not ready for publication. More detailed comments:
- introduction is long and imprecise and it is unclear what the work builds on, all contributions are just mentioned in "absolute" terms instead of in relation to existing work. It only becomes clear on page 3 what the goal/methodology is. Later in the paper, everything seems to build on Boopathy et al. (2023).
- presentation of the work reads like a draft rather than a finished paper: remark about intro/abstract above, but also 3.2 suddenly goes into GP regression without any motivation and then into relatively detailed specifics that seem irrelevant for the method used in the experiments. It makes it very hard to read the paper.
- experiments only compare to the approach by Boopathy and no other methods to compute generalization error/bounds, or quantify inductive bias.

**Questions:**

- can you relate the work to other works than Boopathy in more detail?
- what are main take-aways from the inductive bias definition and the experiments? The results do not seem novel or surprising but maybe I just fail to see their significance.
- is it right that computing the inductive bias definition for neural networks is basically just training $S$ networks from scratch and calculating how many of them generalize well? Isn't that fundamentally the same as evaluating the held-out performance but accounting for randomness due to initialization? It seems inductive bias is simply generalization then.

---

> ### Author Response · Authors · 2023-11-21
>
> Thank you for your valuable comments and constructive feedback.
>
> **Relationship to prior work**
>
> As we mention in Section 2, there are unfortunately few prior works that attempt to quantify inductive biases. As much as possible, we have related our techniques to those established if possible. However, this may not be possible due to the sparsity of research on inductive bias quantification.
>
> We would be happy to relate our work to any other relevant work we may have missed.
>
> **Clarity**
>
> Thank you for your constructive comments. We will closely go through each section of the paper and carefully revise it.
>
> **Main takeaways from paper**
>
> Please see the discussion section for a discussion of the takeaways from this paper. In summary, we highlight three key takeaways:
>
> 1. Higher-dimensional tasks require much greater amounts of inductive bias
> 2. Neural networks encode massive amounts of inductive bias
> 3. Our measure of inductive bias can be used as a tool for interpretability as well as a quantitive guide for task design
>
> **Inductive bias vs. generalization performance**
>
> We would like to clarify that our inductive bias definition measures how much inductive bias is provided by a particular model class in the context of a particular hypothesis space. This is done by converting the generalization performance of a particular model class into an inductive bias measure (Equation 1). When the hypothesis space and model class are both neural networks sets, one way to compute inductive bias is indeed simply to train sample neural networks from the hypothesis space and evaluate the performance of the neural network model class relative to these samples. However, in many cases, evaluating inductive bias within a hypothesis space via sampling may be nontrivial because for large hypothesis spaces, the vast majority of hypotheses may not generalize well. Thus, we propose modeling the test error distribution in Section 3.3, which allows us to estimate the inductive bias without an intractable number of samples.

---

### Official Review · Reviewer_VP96 · 2023-11-08

**Soundness:** 2 fair
**Presentation:** 2 fair
**Contribution:** 3 good
**Rating:** 5
**Confidence:** 3

**Summary:**

The paper proposes a method to directly estimate the "amount of inductive bias" related to a task and a class of hypotheses, which in a nutshell can be summarized as the probability that an hypothesis the class achieves a test loss lower than a given threshold an the given task under a certain probability distribution $p_h$ over hypotheses.
One central challenge of the approach is to model and sample from $p_h$. The authors propose one direct optimization based method and one indirect kernel-based method that hinges on a series of assumptions.
Furthermore, the authors derive bounds for the approximation error of the proposed estimator (where the approximations stems from partial modelling of $p_h$, partial modelling of the test loss distribution and sampling.
The study concludes with two series of experiments, one with the Gaussian RBF hypothesis space and another with neural networks. The authors draw some conclusions regarding the extensive bias encoded by the neural network class and that higher dimensional tasks need (feature?) higher inductive bias.

**Strengths:**

- The problem addressed by the paper is of central importance in the community and I believe that the work could be of some interest because of its novelty
- The paper is mostly well written and easy enough to follow, except for some passages related to the estimation of the test loss distribution

**Weaknesses:**

- For an objective standpoint, I find that the "experimental" claims about 1) high-dimensional tasks -> more inductive bias and especially that 2) "neural networks encode massive amount of inductive bias" rather weak. These are based on an extremely sparse set of experiments and are not backed by theoretical justifications. Especially for 2) the observation even seems to me quite indirect: I do not think that the method proposed can directly quantify "the inductive bias encoded in (some subclasses of?) neural networks. I might have missed some detail, and I would appreciate the authors to highlight what are the supporting evidences for these two claims (and carefully consider their quality).
- From a more subjective standpoint, I have some doubts that the concept of inductive bias can be effectively captured by a (single) scalar number. Rather, the inductive bias of a learning algorithm is the set of explicit and implicit assumption that cause the learning algorithm to "choose" one hypothesis (rather than another) and thereby asserting the way the resulting model generalizes. In this sense, I disagree on the definition of inductive bias that the authors give, although I understand that this somewhat subjective.
- In any case, I find the measure proposed in Eq. 1 rather brittle; by changing threshold and distribution over hypothesis one may obtain essentially arbitrary numbers. Therefore any quantity produced by any estimation procedure of 1 should be heavily contextualized to avoid drawing any unsupported conclusions (this also relates to my 1st concern), casting doubts on the effectiveness and practical use of this measure.
    - on the top of this, I also think that the definition does not capture well the interplay between inductive bias, sample complexity,  hypothesis space and learning algorithm (as an higher-order functional). In particular, different learning algorithms have different sample complexities, which may result in one algorithm performing comparatively better than another up until a certain data regime; e.g. think about few shot-learning. How does (1) captures or addresses this fact? I think (1) could more explicitly incorporate dependence upon a training regime (which could be simply the number of training examples).
   - I also have some doubts about folding the learning algorithm into the probability distribution $p_h$. This causes confusion on how to design $p_h$ (and who should do it) and does not consider important details such as the choice of hyperparameters.
   - the paper would also benefit from a broader and more careful commentary about the introduced measure. How should one interpret the resulting number? What about some notable limit cases (e.g. when $p_h$ is a Dirac delta)
- The assumptions related to kernel-based sampling method seem rather unrealistic (eps. $h(x) = \phi(x) \theta$) and are not well discussed.

Minors:
- Why is $p_\theta$ a Gaussian process (rather than simply a Gaussian distribution?
- $K$ is undefined
- I find that the authors could be clearer about various quantities, e.g. regarding what is a random variable and what is not. For instance, is the training set a random variable or is it considered fixed?

**Questions:**

- How do you define the task dimensionality? Are you referring to a formal definition? if so, which one?
- Why is it necessary to divide the concept of hypothesis space and model class? How is it useful for the derivations and the framework proposed? In any case, I think it would be helpful to introduce a simple running example to help the reader
- How do optimization-based and kernel-based sampling compare?

---

> ### Author Response · Authors · 2023-11-21
> **Response 1/2**
>
> Thank you for your valuable comments and constructive feedback.
>
> **Justification of experimental claims**
>
> We first note that the claim that higher-dimensional tasks require greater inductive bias is not a new finding; indeed, Boopathy et al. find the same result with theoretical backing. Moreover, it is well-known that high-dimensional tasks require more training samples (see Section 2).
>
>
> Next, we argue that neural networks encode massive amounts of inductive bias based on the observation in Figure 4 that the inductive bias provided by particular neural network model classes within a broader neural network hypothesis space is quite small. Thus, given that we already restrict functions to neural networks, the *additional* inductive bias of a particular neural network model class is quite small. Contrast this with the inductive bias in a kernel-based hypothesis space (see Table 1). In this hypothesis space, required inductive biases are orders of magnitude larger. Thus, while the hypothesis space corresponds to kernel-methods does not provide much inductive bias (neural network model classes provide a lot of additional inductive bias), the neural network hypothesis space provides large amounts of inductive bias (neural network model classes within this space provide little additional inductive bias).
>
> **Concerns on the definition of inductive bias**
>
> We note that we do not introduce the idea of quantifying inductive bias with a single number; see Chollet (2019) and Boopathy et al. (2023). Our precise definition of how to quantify inductive bias in Equation 1 is very much in line with these prior works. However, we are happy to change our wording if it would be more consistent with the established literature on inductive bias.
>
> Our inductive bias definition relies heavily on the error threshold and hypothesis distribution *by design*. We believe inductive bias quantifications *must* be placed in the context of a particular hypothesis distribution; for instance, while neural networks may provide large amounts of inductive bias in the context of a relatively large kernel-based hypothesis space, they provide little inductive bias when the hypothesis space is already restricted to only neural networks. We believe it is not meaningful to consider inductive biases quantification without the context of a particular hypothesis space.
>
> We further note that our measure of inductive bias is exactly designed to capture the effect of changing inductive biases such as the learning algorithm under different data regimes (such as few-shot learning vs. big data learning). In the context of Equation 1, this can be quantified by first fixing a common hypothesis distribution $p_h$ under which to compare the different inductive biases. Next, we empirically measure the test set error rates corresponding to different inductive biases under different data regimes. Equation 1 can finally be used to convert these error rates into inductive bias measurements. Thus, we may compare the effect of changing a particular inductive bias (including choice of learning algorithm) under different data regimes.
>
> **Interpreting inductive bias measure**
>
> Please see Section 5 for a discussion on the interpretation of our inductive bias measure. We will add additional details in our revision.
>
> **Assumptions on kernel-based sampling**
>
> The assumption that $h(x) = \phi(x) \theta$ is fundamental to kernel-based sampling; Gaussian processes, for instance, assume that $\theta$ is Gaussian, with $\phi$ setting the kernel of the Gaussian process. We believe that kernel-based hypothesis space is a natural choice for a hypothesis space since, in general, any function has non-zero probability density under a kernel-based hypothesis space. Thus, it is broad enough to capture any model class, while simultaneously allowing for bias toward the most reasonable hypotheses (e.g.. smooth, compositional functions) depending on the choice of kernel.
>
> **Gaussian process notation**
>
> $p_\theta$ is a Gaussian; the resulting distribution over functions is a Gaussian process.
>
> **Clarity**
>
> Thank you for your suggestions on improving the clarity of our submission; we will carefully clarify these points in our revision.
>
> **Task dimensionality definition**
>
> Task dimensionality is defined as the intrinsic dimensionality of the inputs of a task. We will clarify this in our revision.

---

> > ### Author Response · Authors · 2023-11-21
> >
> > **Hypothesis space vs. model class**
> >
> > A hypothesis space corresponds to the space of all functions relevant to consider on a task. A model class is the space of hypotheses used to generalize on a task. Our inductive bias measure quantifies the inductive bias required to generalize to a particular error rate on a task in the context of a particular hypothesis space. It can also be used to quantify the inductive bias provided by a particular model class in the context of the hypothesis space by plugging the error rate of the model class. We use two different terms to refer to these two sets of models to clarify that they play distinct roles in our inductive bias measure.
> >
> > **Optimization-based vs. kernel-based sampling**
> >
> > Kernel-based hypothesis spaces yield much larger inductive biases than optimizing on neural network hypothesis spaces (compare Table 1 to Figure 4). This is because kernel-based hypothesis spaces are much less restricted than neural network hypothesis spaces; neural networks provide many useful smoothness and compositionality constraints that functions sampled from a kernel-based hypothesis space may lack.

---

### Meta-Review · Area_Chair_eUWv · 2023-12-11

**Metareview:**

The submission builds on prior work quantifying the inductive bias needed for a specific threshold of performance on given task. Reviewers had concerns with clarity, originality and significance that prevent the paper from being accepted in its current form.

**Justification For Why Not Higher Score:**

additional work needed in clarity, originality and significance aspects

**Justification For Why Not Lower Score:**

N/A

---

### Decision · Program_Chairs · 2024-01-16

Reject